# Biocompatible Heterogeneous Packaging and Laser-Assisted Fluid Interface Control for In Situ Sensor in Organ-on-a-Chip

**DOI:** 10.3390/mi16010046

**Published:** 2024-12-30

**Authors:** Yu-Hsuan Lin, Shing-Fung Lau, Yen-Pei Lu, Kuo-Cheng Huang, Chien-Fang Ding, Yu-Hsiang Tang, Hsin-Yi Tsai

**Affiliations:** 1Taiwan Instrument Research Institute, National Applied Research Laboratories, Hsinchu 300092, Taiwan; marklin@narlabs.org.tw (Y.-H.L.); ypl@narlabs.org.tw (Y.-P.L.); huangkc@narlabs.org.tw (K.-C.H.); sky520830@narlabs.org.tw (Y.-H.T.); 2Department of Biomechatronics Engineering, National Taiwan University, Taipei 106319, Taiwan; d12631006@ntu.edu.tw (S.-F.L.); cfding@ntu.edu.tw (C.-F.D.)

**Keywords:** organ-on-a-chip (OoC), biocompatible heterogeneous packaging, laser surface modification, wettability, organ twin

## Abstract

The development of bionic organ-on-a-chip technology relies heavily on advancements in in situ sensors and biochip packaging. By integrating precise biological and fluid condition sensing with microfluidics and electronic components, long-term dynamic closed-loop culture systems can be achieved. This study aims to develop biocompatible heterogeneous packaging and laser surface modification techniques to enable the encapsulation of electronic components while minimizing their impact on fluid dynamics. Using a kidney-on-a-chip as a case study, a non-toxic packaging process and fluid interface control methods have been successfully developed. Experimentally, miniature pressure sensors and control circuit boards were encapsulated using parylene-C, a biocompatible material, to isolate biochemical fluids from electronic components. Ultraviolet laser processing was employed to fabricate structures on parylene-C. The results demonstrate that through precise control of processing parameters, the wettability of the material can be tuned freely within a contact angle range of 60° to 110°. Morphological observations and MTT assays confirmed that the material and the processing methods do not induce cytotoxicity. This technology will facilitate the packaging of various miniature electronic components and biochips in the future. Furthermore, laser processing enables rapid and precise control of interface conditions across different regions within the chip, demonstrating a high potential for customized mass production of biochips. The proposed innovations provide a solution for in situ sensing in organ-on-a-chip systems and advanced biochip packaging. We believe that the development of this technology is a critical step toward realizing the concept of “organ twin”.

## 1. Introduction

Organ-on-a-chip is an advanced type of biochip designed to simulate the functions and physiological environments of human organs [1,2,3,4,5,6,7,8,9,10,11]. It has potential applications in precision medicine, toxicological research, and other areas, significantly contributing to the reduction of animal testing. The development of organ-on-a-chip technology requires extensive microengineering techniques, which are primarily employed to construct “three-dimensional cell culture chambers, microfluidic channels, and sensing electronic components”. Among these, microfluidic channels serve as a microenvironment that mimics the transport of bodily fluids, such as blood and interstitial fluid, within the human body. These channels can replicate the complex interactions within organs. The microfluidic channels of the chip must be fabricated using biocompatible polymer materials. Their structures need to be highly precise to effectively control the flow and delivery of fluids. The primary distinction between organ-on-a-chip and traditional biochips lies in its capability for “dynamic culture” [12,13,14,15,16]. This feature enables the continuous supply of nutrients and oxygen to three-dimensional cell structures while removing metabolic waste over an extended period. Such functionality closely approximates the natural physiological conditions within the human body. Currently, organ-on-a-chip technology faces several major challenges, including three-dimensional organ cell culture, the completeness of nutrient and metabolite supply through microfluidics, more compact in situ sensors, the biocompatibility of electronic components within the chip, and more precise microfluidic control. Reproducing the behavior of human organs is an exceedingly complex endeavor. Achieving more biomimetic designs and manufacturing processes is essential, and it remains the ongoing pursuit of all bioengineers.

Three-dimensional (3D) cell culture can more realistically mimic the structural arrangement within organs, such as blood vessels, alveoli, or glomeruli. Compared to traditional two-dimensional (2D) culture systems, 3D systems promote natural cell differentiation and functional expression [17,18,19]. This approach requires more complex and precise support for nutrient delivery and waste metabolism, bringing the chip closer to the characteristics of real organs. These tasks typically fall under the expertise of biochemists, who specialize in cultivating and observing cellular states. On the other hand, electromechanical engineers focus on the biomimetic fabrication of chips, which can be regarded as the “digital organ twin” of the bioengineering field [20,21,22]. The development of in situ sensors is one of the most critical aspects. Adequate and accurate sensors enable real-time monitoring of key biological indicators, such as cellular metabolism, oxygen consumption, or bioelectric signals. These sensors must be sufficiently small and positioned near the cells to obtain accurate biological data in an in situ manner [23,24,25,26,27]. Long-term in situ dynamic monitoring can provide valuable insights into complex physiological processes. However, sensors are often electronic components, raising concerns about their biocompatibility. Toxic components may interfere with normal cell behavior, leading to inaccurate experimental results. Furthermore, the heterogeneous structure of sensor surfaces may alter fluid conditions within the chip, causing discontinuities in viscosity and shear forces within microfluidic channels. Such inconsistencies can compromise the accuracy and stability of the simulated physiological environment. The risks associated with misleading experimental results, particularly in terms of safety, cannot be ignored. Traditionally, these issues might have been overlooked. However, as organ-on-a-chip technology advances, we believe that more precise engineering techniques are essential to ensure reliability and accuracy.

Since in situ sensors are electronic components, they are typically unsuitable for direct contact with cultured cells or with the culture medium. It is essential to use high-quality biocompatible materials for encapsulation. Suitable materials include polydimethylsiloxane (PDMS), polyimide (PI), polyether ether ketone (PEEK), silicon dioxide (SiO₂), and parylene-C (Pa-C). Each of these materials offers specific advantages, such as isolating biological fluids from electric circuits or resistance to corrosion and high temperatures. Their common characteristic is excellent biocompatibility. Thin film is an effective encapsulation method, as it ensures minimal thickness to avoid reducing signal sensitivity. Signals can penetrate the film, while fluids and cells are prevented from coming into direct contact with the sensor. However, biocompatible encapsulation alone is insufficient. The isolation layer on the sensor surface may differ from the fluid dynamics conditions within the microfluidic channels. Therefore, hydrophobic or hydrophilic surface modifications on the encapsulation layer are necessary to adjust the surface properties of the sensor to meet specific requirements. Within an organ-on-a-chip system, the hydrophilic or hydrophobic nature of each region should be adjusted based on its functional needs. Generally, microchannels that deliver nutrients and gases should be as hydrophobic as possible to reduce contamination accumulation. Conversely, areas near organ cells may need to be hydrophilic to facilitate biochemical reactions and increase reaction surface area. The sensor’s hydrophilic or hydrophobic treatment depends on its location: hydrophobic surfaces enhance long-term stability, while hydrophilic surfaces promote cell adhesion and accurate in situ sensing. Thus, establishing surface modification techniques that allow for adjustable hydrophilic and hydrophobic properties on biocompatible encapsulation materials is crucial. Common methods include plasma treatment, chemical vapor deposition (CVD), electrochemical methods, and laser surface treatment. Among these, laser surface treatment is highly recommended due to its ability to rapidly create different surface conditions in various regions, significantly reducing manufacturing costs [28,29,30].

In this study, a biocompatible heterogeneous encapsulation and surface modification technique was developed and applied to a kidney organ-on-a-chip. The kidney is a critical metabolic organ responsible for waste filtration in the human body, yet it is highly susceptible to damage. The development of kidney organ-on-a-chip technology facilitates the study of the effects of drug toxicity and hypertension on kidney function. Drug toxicity can be indirectly assessed by analyzing the presence of foreign substances in the filtrate, while hypertension requires in situ sensors to monitor the actual pressure exerted on glomerular cells. As shown in Figure 1, the kidney organ-on-a-chip consists of two layers of microfluidic channels, as depicted in the cross-sectional diagram. The upper layer simulates blood circulation, while the lower layer replicates urine flow. Between these layers are three types of kidney cells cultured in layers: endothelial cells, a porous membrane, and podocytes. The blood circulation mimics the afferent arteriole, while the urine flow replicates excretory behavior before reaching the proximal tubule. The kidney cells in the middle demonstrate glomerular filtration functionality under pressure exerted by the microfluidic channels. By increasing the flow rate in the upper fluidic layer, filtration pressure can be enhanced. To accurately measure the in situ pressure exerted on the kidney cells, miniature pressure sensors must be placed adjacent to the cells. As previously discussed, to ensure the biocompatibility of electronic components and achieve the desired surface characteristics, biocompatible heterogeneous encapsulation and surface modification must be precisely implemented. In this study, miniature in situ pressure sensors were coated with parylene-C to maintain biocompatibility. Using laser surface modification, various structures were engraved on the parylene-C layer. The results demonstrated that different hydrophilic and hydrophobic surface conditions could be freely established on the sensor. These surface structures did not compromise the material’s biocompatibility, as confirmed by ISO 10993 [31,32] testing. This indicates that encapsulating various electronic components within biochips without affecting fluid properties is feasible. The research findings contribute to the advancement of in situ sensor technology within organ-on-a-chip systems. The rapid and penetrative capabilities of laser optics highlight its potential for cost-effective and localized processing. Laser surface modification can minimize interference with the living environment of 3D cell cultures. The development of this technology will benefit not only organ-on-a-chip systems but also other microbioengineering advancements.

## 2. Working Principle and Experimental Setup

The kidney organ-on-a-chip consists of a symmetrical structure with upper and lower layers, as shown on the left of Figure 2 (only the lower layer is illustrated). Through injection molding, manufacturing costs can be significantly reduced. A pressure sensor (green object) is embedded in the middle region of the chip, near the kidney cells, allowing it to be classified as in situ. The pressure detected by the sensor closely approximates the pressure experienced by the glomerulus. The sensor is made of piezoelectric material and is soldered onto a miniature circuit board, with signals output from the side of the chip. In humans, the glomerular blood hydrostatic pressure (GBHP) in the afferent arterioles is approximately 55 mmHg, Bowman’s capsular hydrostatic pressure (BCHP) is around 15 mmHg, and colloidal osmotic pressure (COP) is about 30 mmHg. Thus, the standard net filtration pressure (NFP) in humans is approximately 10 mmHg. To develop a system capable of detecting the effects of hypertension on the kidney, the pressure sensor is designed to measure within a range of 0~215 mmHg, with a resolution of 0.75 mmHg. This enables incremental pressure increases above the standard NFP. By observing albumin leakage, the damage to podocytes in the kidney can be evaluated. The biochip’s structure is manufactured with high precision, and testing confirms that there is no leakage within the microfluidic channels. The pressure sensor and its circuit board are extremely small. As shown in the right of Figure 2, two developed in situ pressure sensors are compared with a Taiwanese one-dollar coin. The coin has a diameter of 20 mm, while the sensors measure only 2 mm in both length and width. The miniature size of the sensors facilitates the integration of additional sensing capabilities within microchips.

To achieve biocompatible heterogeneous encapsulation for in situ sensors, both the pressure sensor and the circuit board were coated with parylene-C. Parylene-C, a polymer derived from chlorinated para-xylylene, features excellent biocompatibility, corrosion resistance, gas impermeability, and uniform thin-film properties. It is ideal for encapsulating biochips and implantable devices, providing superior protection and reliable isolation between sensitive electronics and biological tissues, ensuring stability and durability in biomedical and microelectronic applications. This thin film has a thickness of approximately 8 μm and completely isolates fluids and conductivity by covering FR-4 (fiberglass epoxy resin), polycarbonate, and soldered metal. This ensures the long-term stable operation of both biological cells and sensors. As shown in Figure 3a, 355 nm solid-state ultraviolet lasers were used for surface modification of parylene-C. The processing depth was controlled to avoid penetrating through the film. The top view, as shown in Figure 3b, reveals the green areas representing the circuit board and the brown areas representing the pressure sensor. Laser processing created various surface structures on parylene-C to achieve different hydrophilic and hydrophobic characteristics. Grid structures and circular arrays were selected as the pattern for testing, as prior research has demonstrated their effectiveness as candidates [33,34]. Prior to conducting the experiments, we initially explored a suitable range of processing parameters for polymer materials. The finalized results indicated the need for lower laser power, along with appropriate frequency and scanning rate. Variations in structures and processing parameters aid in analyzing the hydrophilic and hydrophobic properties of the material surface. In grid structures, line width and pitch were selected as variables, while in circular arrays, circle diameter and pitch were used. After processing the various structures, a confocal optical microscope (Keyence Corporation, Osaka, Japan, VK-X2000) was employed to observe the surface morphology and calculate roughness. A contact angle measurement system (First Ten Ångstroms, Inc., Newark, CA, USA, FTA 188) was used to measure the contact angle between the solid and liquid interface. Surfaces with a contact angle of less than 90° were classified as hydrophilic, while those with an angle greater than 90° were classified as hydrophobic.

## 3. Experimental Results and Discussion

The contact angle of parylene-C with liquids typically ranges between 80° and 100°, with slight variations depending on the liquid and coating conditions. In this study, the parylene-C layer on the sensor surface initially exhibited a contact angle of approximately 88.5°, indicating slight hydrophilicity. Traditionally, surface energy is increased through oxygen plasma or ozone treatments, enhancing hydrophilicity. Conversely, surface energy is reduced through surface roughening via laser or etching, enhancing hydrophobicity. Laser processing, with its advantages in speed, spatial selectivity, and cost efficiency, was adopted in this study for surface modification. In this study, we explored the control of hydrophilic and hydrophobic properties on parylene-C through the fabrication of grid structures and circular arrays. Additionally, various laser processing parameters were tested to evaluate their gradual effects on surface properties. This high tunability allows for flexible adjustment of liquid flow characteristics across different regions of the sensor or microfluidic channels. The laser used had a power of 1 W, a repetition frequency of 100 kHz, and a scan rate of 0.1 m/s.

Figure 4 shows the results of the grid structure processing. From left to right, the figure displays line width and pitch, the appearance of the processed structure, confocal 3D optical imaging, and liquid contact angle measurements. The line width was fixed at 0.06 mm, while the pitch values were 0.05, 0.08, 0.11, and 0.17 mm. The structural appearance shows differences in square pattern size and density. The 3D optical images reveal the height information of the processed structures, with an average depth of approximately 11 μm. The far-right shows variations in contact angle, where the presence of square structures increases the hydrophobicity of parylene-C, resulting in contact angles all greater than 100°. Figure 5 presents the numerical analysis results. The gray horizontal line represents the hydrophilic–hydrophobic boundary (contact angle = 90°), while the purple dashed line indicates the original hydrophilic condition of the parylene-C film. It can be observed that as the pitch increases, the hydrophobicity initially rises but does not continue indefinitely. After reaching a peak, it begins to decline. The maximum contact angle of 113.6° was observed at a pitch of approximately 0.11 mm. Each value in the graph represents the average of three measurements taken from different areas. Due to the unevenness of the sensor substrate and the parylene-C material, as well as errors introduced by laser processing, the surface roughness (Ra) measurements may not have a direct correlation and should be considered as reference data only.

Figure 6 presents the results of circular array fabrication. The laser settings were consistent with the previous configurations (1 W, 100 kHz, 0.1 m/s). From left to right, the figure displays the following: the diameter and pitch of the circles, the processed structural appearance, confocal 3D optical imaging, and liquid contact angle measurements. The circular diameter was fixed at 0.1 mm, while the pitch values were set to 0.3, 0.45, 0.6, and 0.75 mm. Parylene-C, a polymer thin-film material, has a low thermal conductivity (~0.12 W/m·K) and high ultraviolet absorption. When exposed to laser energy, it undergoes melting and vaporization. As a result, slight deformations of the circular structures can be observed in the processed structural appearance. With increasing pitch, the holes become more dispersed and less frequent. The 3D optical imaging provides height information for the processed structures, with an average depth of approximately 93 μm. The rightmost section of Figure 6 illustrates the variations in contact angle. The presence of circular structures induces hydrophilicity on the parylene-C surface; however, the effect is relatively limited. The contact angles primarily fall within the range of 80° to 90°. Compared to oxygen plasma or ozone treatments, the hydrophilicity enhancement induced by these structures is minimal, though still measurable. Figure 7 provides numerical analysis results. The gray horizontal line represents the boundary between hydrophilic and hydrophobic surfaces (contact angle = 90°), while the blue dashed line indicates the original hydrophilic condition of the parylene-C film. Opposite trends are observed between circular arrays and grid structures. As the pitch increases, hydrophilicity initially improves but does not persist indefinitely. After reaching a peak, the trend reverses. The lowest contact angle, 82.7°, occurs at a pitch of approximately 0.6 mm. Each value in the graph still represents the average of three measurements from different regions. These results demonstrate that circular arrays on parylene-C provide only limited enhancement of hydrophilicity.

To analyze the effects of additional structural variations on hydrophilicity and hydrophobicity, different laser scanning speeds were incorporated into the experiment. Figure 8 shows the results of grid structure fabrication at various scan rates. The fixed parameters included the following: laser power at 1 W, repetition frequency at 100 kHz, and line width of 0.06 mm. The pitch was set to 0.11 mm for this experiment, as this parameter demonstrated the highest hydrophobicity (contact angle of 113.6°) in Figure 5. The laser scan rate was varied under these parameters to investigate changes in the material’s surface properties. The laser scan rates were set to 0.01, 0.1, 0.2, 0.3, 0.4, and 0.5 m/s. Figure 8 shows that extremely slow laser processing speeds increase hydrophilicity, as indicated by reduced contact angles. It is estimated that a scan rate below 0.05 m/s would result in contact angles dropping below 90°, indicating hydrophilic surfaces. At scan rates between approximately 0.05 and 0.3 m/s, the parylene-C surface exhibited hydrophobicity, with a peak hydrophobicity observed at a scan rate of around 0.1 m/s. Beyond this rate, hydrophobicity gradually diminished, and the surface returned to hydrophilic behavior (contact angles below 90°) when speeds exceeded 0.3 m/s. At a scanning speed of 0.5 m/s, the contact angle was approximately 60°. The enhancement of hydrophilicity was remarkably significant. The measured surface roughness (Ra) trends corresponded with the changes in contact angle, which is a satisfactory outcome. These results indicate that by varying the laser processing parameters during grid structure fabrication on parylene-C, it is possible to precisely tune the surface hydrophilicity and hydrophobicity within a range of 60° to 110°. This aligns with our expectation that high-speed laser processing can efficiently and dynamically adjust liquid flow characteristics across different regions of sensors or microfluidic channels.

Figure 9 illustrates the liquid contact angle images under several representative laser processing parameters. Figure 9a shows the original slightly hydrophilic property of the parylene-C surface. Prior to laser processing, the contact angle was 89°, which can be considered a near-neutral surface. It can be understood that parylene-C, with its high biocompatibility and neutral characteristics, is highly suitable as a material for biochip encapsulation. Figure 9b presents the most hydrophobic result from the experiment detailed in Figure 8, with a contact angle of approximately 111°. This indicates relatively weak interactions between the surface and the liquid, promoting the liquid’s tendency to maintain a spherical shape. Such behavior helps minimize interference from channel materials on the culture medium and cells. With changes in laser processing conditions, a balance between micro/nanostructures and material surface energy can be achieved. As shown in Figure 9c, the initially hydrophobic surface gradually returns to the neutral interface of the original material. This is attributed to increased laser processing speeds, which reduce surface roughness and subsequently enhance hydrophilicity. Figure 9d demonstrates a surface where liquid readily spreads, with stronger interactions between the surface energy and the liquid. The contact angle was approximately 61°, a property well-suited for sensors and regions near cells to enhance interactions such as reaction area and biochemical activity. As a supplementary note, after laser processing at varying rates, the circular array demonstrated a less effective range of expandable contact angles. Therefore, the results of this experiment are not included in this paper.

The wettability of parylene-C surfaces with patterned microstructures can be explained through surface energy principles and roughness-induced wetting models, specifically the Wenzel and Cassie–Baxter models. Surface roughness significantly influences the apparent contact angle by altering the solid–liquid interface. In the Wenzel model, the liquid penetrates the microstructures, increasing the effective contact area and enhancing wettability for inherently hydrophilic surfaces. The apparent contact angle ($\theta_{w}$) follows the relationship $\cos\theta_{w}=r\cos\theta_{Y}$, where r > 1 represents the surface roughness factor and $\theta_{Y}$ denotes the intrinsic (Young’s) contact angle of the smooth surface. Conversely, the Cassie–Baxter model describes a composite interface where the liquid droplet rests partially on trapped air pockets within the microstructure, minimizing solid–liquid contact. The apparent contact angle ($\theta_{C}$) is given by $\cos\theta_{C}=f\cos\theta_{Y}+(1-f)$, where f denotes the fraction of solid–liquid contact. This mechanism effectively increases hydrophobicity for inherently hydrophobic surfaces. Experimental observations show that grid-like structures on parylene-C enhance hydrophobicity due to air entrapment, whereas circular arrays increase hydrophilicity by promoting liquid infiltration. These findings demonstrate that microstructural geometry and surface energy collectively modulate wettability for tailored functional applications.

Parylene-C is inherently highly biocompatible; however, laser surface modification may potentially introduce toxicity due to phenomena such as oxidation and charring. This possibility must be experimentally verified to ensure the practicality and usability of this study. To address this, the processed samples were evaluated according to the ISO 10993 standard validation procedures. Generally, cytotoxicity testing can be conducted using two approaches: one based on ISO-standard cells and the other using specific cultured cells. In this study, we chose to follow ISO 10993-5 (in vitro cytotoxicity testing) to ensure compliance with regulatory requirements. Accordingly, L929 mouse fibroblast cells were selected for the test. This approach offers both advantages and limitations. The ISO standard is internationally recognized, providing robust evidence of safety and regulatory compliance. However, it may not fully reflect the actual cytotoxicity in specific target cells (e.g., kidney cells). We selected the standard method considering the potential applicability of this study to various organ-on-a-chip platforms. The kidney organ-on-a-chip mentioned in the introduction serves merely as an example to illustrate the need for sensor encapsulation technologies. Looking ahead, as multiple organ-on-a-chip systems (e.g., heart, liver, and kidney) may eventually be interconnected, a general biocompatibility standard that ensures non-cytotoxicity is essential. Balancing these considerations, we have decided to adopt the ISO 10993 standard procedure for biocompatibility verification in this study as a temporary measure. The selection of cells and reagents is detailed as follows: cytocompatibility was evaluated using NCTC clone 929 (L929) cell lines obtained from FIRDI (Food Industry Research and Development Institute, Taiwan). The L929 cells were cultured in Eagle’s Minimum Essential Medium (MEM) (Gibco, Grand Island, New York, USA), supplemented with 10% (*v*/*v*) fetal bovine serum (FBS) (Gibco, USA). The 3-(4,5-Dimethylthiazol-2-yl)-2,5-diphenyltetrazolium bromide (MTT) reagent was purchased from Gibco (USA) for in vitro cytotoxicity testing. Isopropanol and dimethyl sulfoxide (DMSO) were purchased from Sigma-Aldrich (St. Louis, MO, USA), and PBS buffer was obtained from Gibco (USA). For the kidney organ-on-a-chip in situ sensor, surface-modified parylene-C coatings were subjected to laser processing, after which their extracts were prepared. The biochip and sensor were sterilized under UV light for 24 h before extraction. Extracts were prepared using Minimum Essential Medium (MEM) supplemented with 10% fetal bovine serum (FBS) at 37 °C for 24 h, maintaining a surface area-to-volume ratio of 3 cm /mL. After extraction, the solution was collected and used for cytotoxicity evaluation via the MTT assay.

Next, cytotoxicity evaluation was conducted. The cytotoxicity of the biochip and sensor was assessed using the MTT assay. L929 cells were seeded into 96-well culture plates at a density of 1 × 10⁴ cells per well and incubated at 37 °C for 24 h. Following incubation, the culture medium was replaced with the prepared extracts. Three controls were included: a blank control containing medium with 10% FBS, a positive control containing culture medium with 10% DMSO (toxic to cells), and a negative control containing culture medium with 10% PBS (non-toxic to cells). All groups were analyzed in triplicate. Following 24 h of incubation with the test and control groups, the medium was replaced with 50 μL of MTT solution (1 mg/mL), and the cells were incubated for an additional 2 h at 37 °C. The resulting formazan crystals were dissolved by adding 100 μL of isopropanol to each well, followed by a 10-min incubation. The optical density (OD) at 570 nm was measured using a Microplate Reader (SpectraMax^®^ M2e, Molecular Devices, San Jose, CA, USA). The cell viability percentage was calculated using the following formula:$$\;\;\;\;\;\;\;\;\;viability\;percentage=\left(\frac{OD570\;nm\;of\;sample}{OD570\;nm\;of\;blank\;control}\right)\times 100\%$$

Cytotoxicity was determined based on cell viability: if viability was reduced to less than 70% of the blank control, the sample was considered to have cytotoxic potential.

The next step involved conducting the cytotoxicity assay: morphological observation and MTT analysis. As shown in Figure 10, cell morphology was observed under a microscope to assess the effects of the extracts. In the positive control group, more than 70% of the cell layers exhibited rounded or lysed cells, with partial destruction of the cell layers and more than 50% growth inhibition. In contrast, the extract sample group, the negative control group, and the blank control group displayed similar cell morphology. Cells in these groups showed discrete intracytoplasmic granules with no signs of cell lysis or growth inhibition. The MTT assay further validated these observations. The results of the MTT assay for cytotoxicity are shown in Figure 11. The blue bars represent groups without waterproof treatment, while the green bars represent groups with waterproof treatment. From left to right, the groups are as follows: no structure, 0.6 circular hole structure, 0.3 circular hole structure, 0.17 cross-hatch structure, 0.11 cross-hatch structure, PC: positive control group, and NC: negative control group. Positive control (PC) means there must be cytotoxicity, and negative control (NC) means there is no cytotoxicity. The viability percentage of L929 cells exposed to biochip/sensor extracts exceeded 70% in all tested samples. According to standard criteria, this indicates that the laser-fabricated biochip/sensor meets the requirements for non-cytotoxic biomaterials. These findings collectively demonstrate that laser-modified parylene-C materials are biocompatible and suitable for biomedical applications.

## 4. Conclusions

Bionic organ-on-a-chip technology enables more precise simulation of human organ functions and physiological environments. Among the critical aspects are in situ sensors and biochip encapsulation techniques. This study aims to develop biocompatible heterogeneous encapsulation and surface modification techniques. Using a kidney-on-a-chip as an implementation example, miniature pressure sensors embedded within the chip were encapsulated with parylene-C. This material established a biocompatible interface for the microfluidic channels. The encapsulated sensors were isolated from water and electricity, thereby extending their operational lifespan. In this research, ultraviolet laser processing was employed to create various structures on parylene-C, enabling observation of their effects on hydrophilicity and hydrophobicity. The experimental results revealed that grid structures enhanced hydrophobicity, achieving a maximum contact angle of 113.6°, while circular arrays increased hydrophilicity, with a minimum contact angle of 82.7°. Without any processing, the material exhibited near-neutral properties with a contact angle of 89°. We believe that further fine-tuning of laser parameters, such as power and frequency, could slightly enhance the interface characteristics produced by these structures. Additionally, experiments involving varying laser scan rates were conducted. The grid structure demonstrated a broader range of contact angle control under different scan rates, with hydrophilicity and hydrophobicity ranging from 60° to 110°. Importantly, laser-processed parylene-C maintained high biocompatibility. Morphological observations and MTT assays showed that the viability of L929 cells exposed to the interface extracts exceeded 70%, meeting the criteria for non-cytotoxic biomaterials. The findings demonstrate that integrating various electronic components into biochips while adjusting fluidic interface parameters is feasible. Moreover, performance can be further improved through fine-tuning of processing parameters. The rapidity, high spatial selectivity, and penetrative properties of laser optics demonstrate advantages in localized processing and cost efficiency. We believe this study offers a novel approach to the field of bioelectronic component integration and encapsulation.

## Figures and Tables

**Figure 1 micromachines-16-00046-f001:**
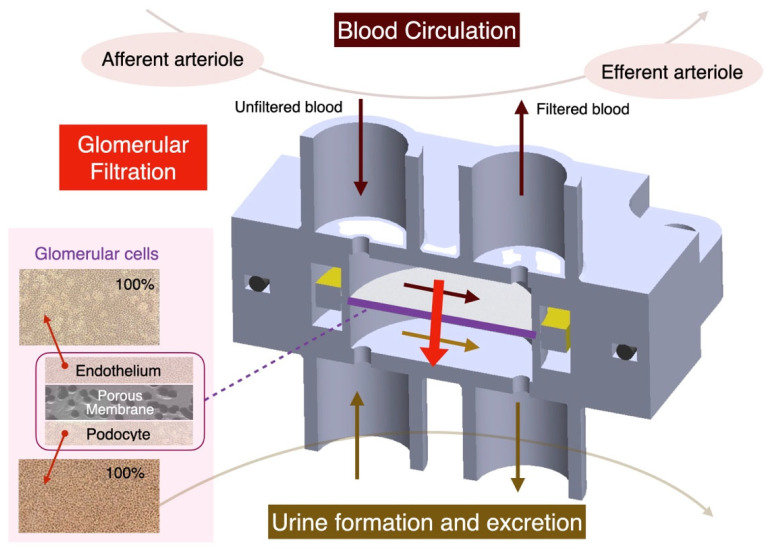
Kidney OoC: platform for advancing/developing technologies.

**Figure 2 micromachines-16-00046-f002:**
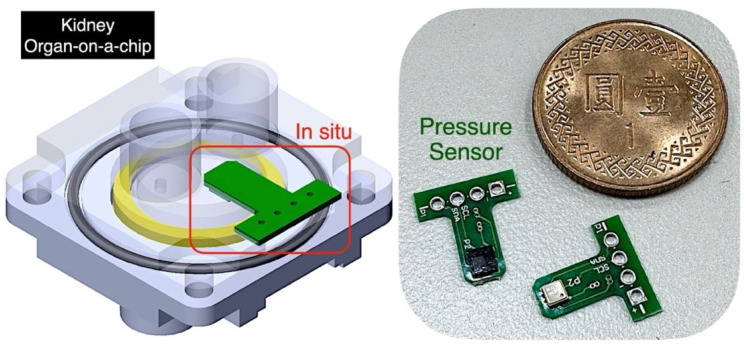
Kidney OoC structure and miniature in situ pressure sensors.

**Figure 3 micromachines-16-00046-f003:**
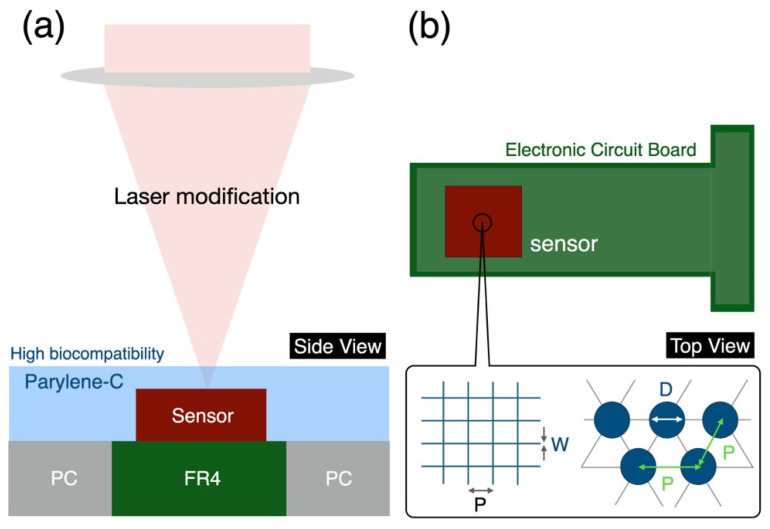
(**a**) Side view showing laser processing on a sensor with a biocompatible layer. (**b**) Top view showing various structures fabricated on a biocompatible material.

**Figure 4 micromachines-16-00046-f004:**
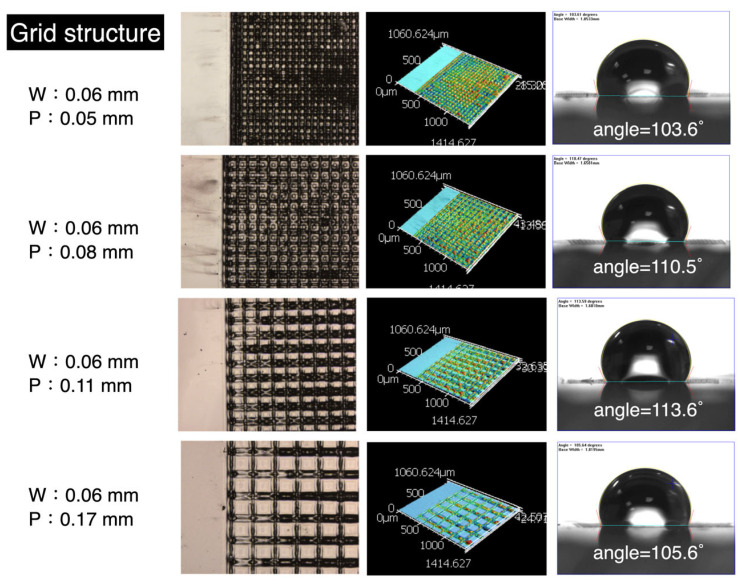
Grid structure results from laser processing on parylene-C.

**Figure 5 micromachines-16-00046-f005:**
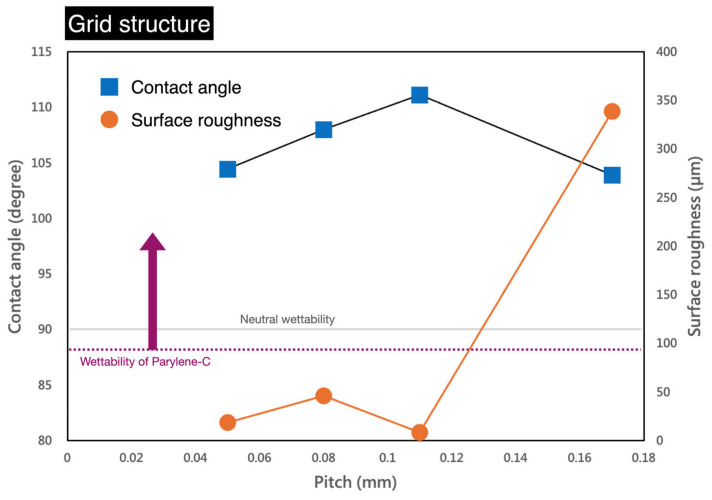
Numerical analysis of contact angles for varying grid structure pitches.

**Figure 6 micromachines-16-00046-f006:**
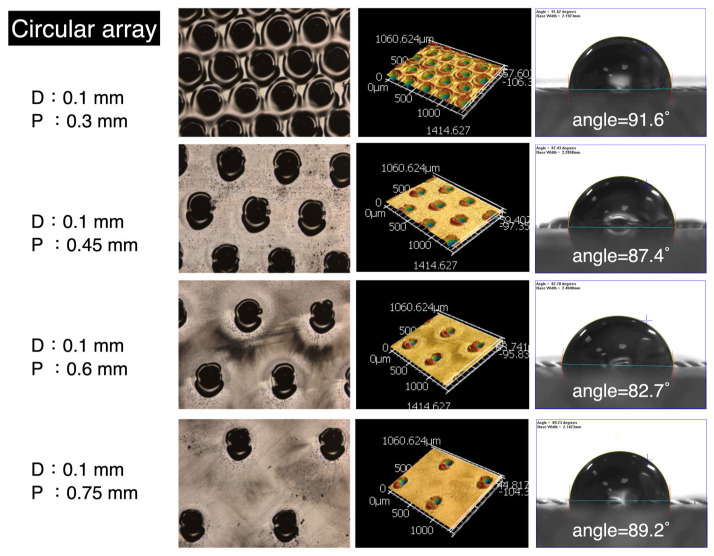
Circular structure results from laser processing on parylene-C.

**Figure 7 micromachines-16-00046-f007:**
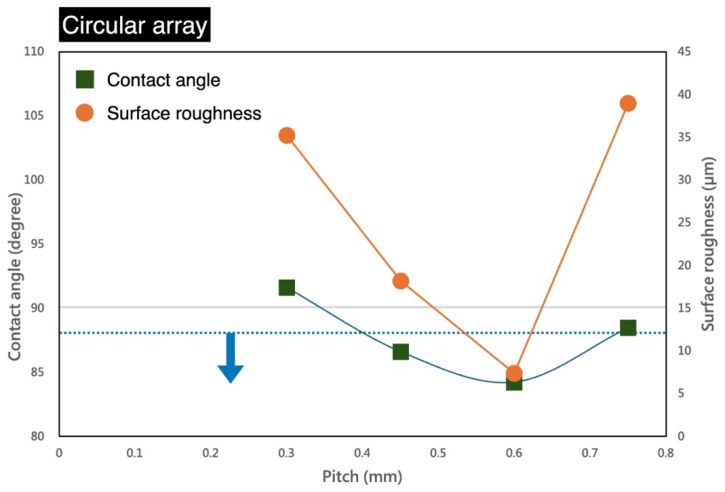
Numerical analysis of contact angles for varying circular structure pitches.

**Figure 8 micromachines-16-00046-f008:**
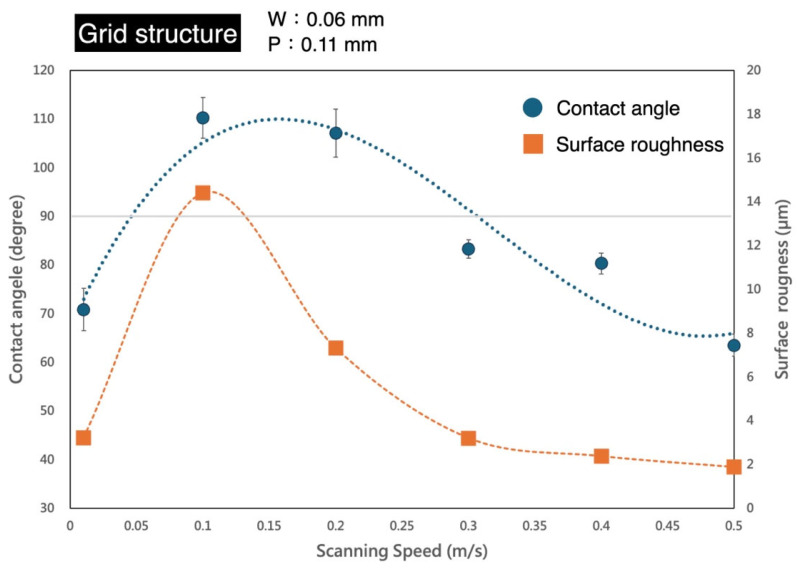
Contact angle analysis of grid structures under varying laser scan rates.

**Figure 9 micromachines-16-00046-f009:**
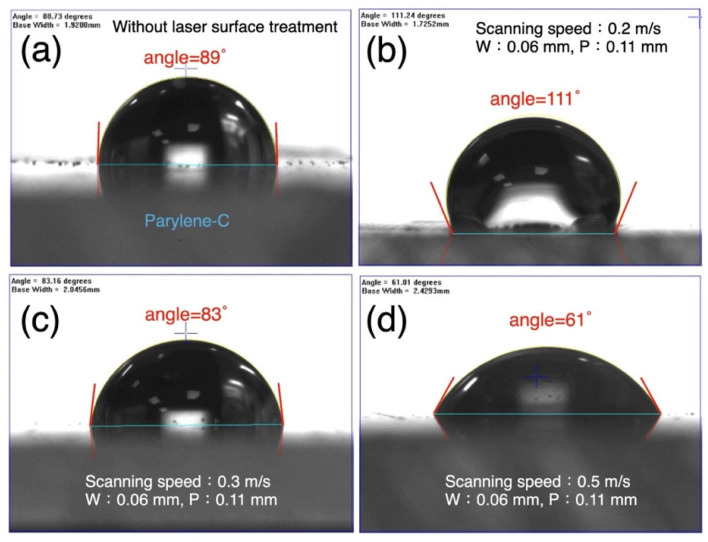
Controlling hydrophilicity and hydrophobicity through laser processing rates. (**a**) Untreated (89°), (**b**) 0.2 m/s (111°), (**c**) 0.3 m/s (83°), and (**d**) 0.5 m/s (61°).

**Figure 10 micromachines-16-00046-f010:**
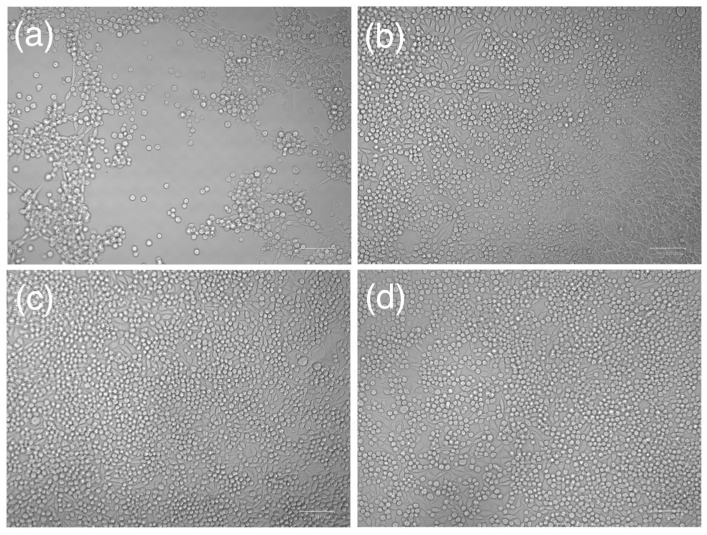
Cell morphology observed 24 h after cell incubated with sample extracts or control medium: (**a**) positive control with 10% DMSO, (**b**) negative control with 10% PBS, (**c**) blank control, and (**d**) Bioship sample extracts.

**Figure 11 micromachines-16-00046-f011:**
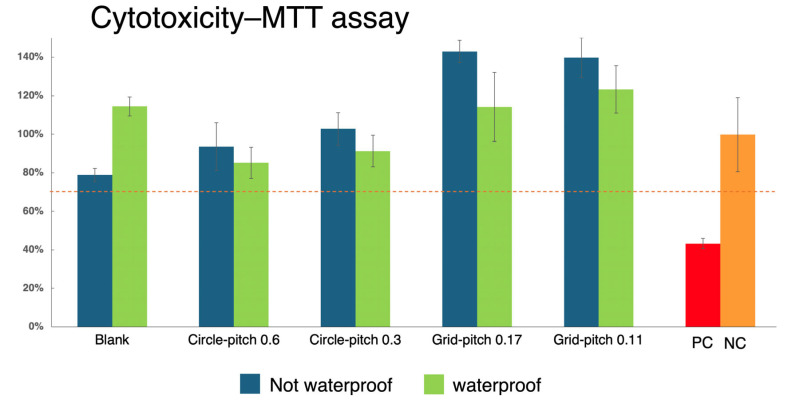
Results of cytotoxicity assay morphological observation and MTT analysis.

## Data Availability

The original contributions presented in this study are included in the article. Further inquiries can be directed to the corresponding author.

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
