# Peer review of "Biocompatible Heterogeneous Packaging and Laser-Assisted Fluid Interface Control for In Situ Sensor in Organ-on-a-Chip"

_micromachines, 2024, doi:10.3390/mi16010046_

Round 1

Reviewer 1 Report

Comments and Suggestions for Authors

The manuscript primarily introduces a study on biocompatible heterogeneous packaging and laser-assisted fluid interface control for Organ-on-a-Chip (OoC) technology. Using a kidney chip as a case study, the research explores the packaging methods for micro pressure sensors and control circuit boards, and modifies the surface of Parylene-C material through ultraviolet laser processing to adjust its hydrophilicity and hydrophobicity. The experimental results show that by precisely controlling the laser processing parameters, the wettability of the material can be freely adjusted within a contact angle range of 60° to 110°. Additionally, morphological observations and MTT cytotoxicity tests confirm that the material and its processing method are non-cytotoxic. This technology provides solutions for the packaging of various microelectronic components and biochips, and demonstrates the potential of laser processing in customized large-scale production of biochips. The reviewer recommends minor revisions and offers the following suggestions:

1.       The study uses multiple laser processing parameters (such as power, frequency, scanning speed), but does not provide detailed explanations for their selection. It is suggested to add an explanation in the text about why these specific parameter ranges were chosen for the experiments.

2.       While Parylene-C is a commonly used biocompatible material, its basic properties are described briefly in the paper. It is recommended to provide a more detailed introduction to Parylene-C, including its physicochemical properties and the reasons for selecting it as the encapsulation layer.

3.       The text mentions that grid structures enhance hydrophobicity while circular arrays enhance hydrophilicity, but it does not delve into the reasons behind these differences in wettability. It is recommended to include an analysis of the microscopic morphology of these structures and their impact on surface energy in the discussion section to help readers understand the underlying mechanisms.

4.       The current cytotoxicity testing is based on a 24-hour MTT experiment. It is suggested to include longer-term cytotoxicity assessments (such as 7 days or 14 days) to verify the safety of the material over extended use.

5.       L929 cells are a commonly used model for cytotoxicity testing, but to better simulate actual application environments, it is recommended to test with additional cell types (such as kidney cells or other organ-specific cells) to ensure the material's broad applicability.

6.       Some important works on the use of sensors in organ-on-a-chip devices need to be cited such as:

Optical sensor: Guo Z, Yang CT, Chien CC, Selth LA, Bagnaninchi PO, Thierry B. Optical Cellular Micromotion: A New Paradigm to Measure Tumor Cells Invasion within Gels Mimicking the 3D Tumor Environments. Small Methods. 2022 e2200471. doi: 10.1002/smtd.202200471.

Electrical sensor: Aydogmus, H., Hu, M., Ivancevic, L. et al. An organ-on-chip device with integrated charge sensors and recording microelectrodes. Sci Rep 13, 8062 (2023). https://doi.org/10.1038/s41598-023-34786-5

7.       Some paragraphs contain complex language. It is suggested to simplify the expression to make the article more readable. For example, long sentences can be reduced, and simpler vocabulary can be used to convey information.

8.       The symbols and units used in the text are sometimes inconsistent (e.g., the unit for contact angle is sometimes "°" and sometimes omitted). It is suggested to standardize the use of symbols and units throughout the text to ensure consistency.

Author Response

Reviewer’s comment,

  • Reviewer#1:

The manuscript primarily introduces a study on biocompatible heterogeneous packaging and laser-assisted fluid interface control for Organ-on-a-Chip (OoC) technology. Using a kidney chip as a case study, the research explores the packaging methods for micro pressure sensors and control circuit boards, and modifies the surface of Parylene-C material through ultraviolet laser processing to adjust its hydrophilicity and hydrophobicity. The experimental results show that by precisely controlling the laser processing parameters, the wettability of the material can be freely adjusted within a contact angle range of 60° to 110°. Additionally, morphological observations and MTT cytotoxicity tests confirm that the material and its processing method are non-cytotoxic. This technology provides solutions for the packaging of various microelectronic components and biochips, and demonstrates the potential of laser processing in customized large-scale production of biochips. The reviewer recommends minor revisions and offers the following suggestions:

  1. The study uses multiple laser processing parameters (such as power, frequency, scanning speed), but does not provide detailed explanations for their selection. It is suggested to add an explanation in the text about why these specific parameter ranges were chosen for the experiments.

Response: We appreciate the reviewer’s suggestions. In the revised manuscript, we have added a brief explanation regarding the selection of laser processing parameters and cited relevant previous studies. The laser processing parameters were tested and fine-tuned to better suit polymer materials, with a primary focus on reducing power.  

Revised in 176-181:

  1. While Parylene-C is a commonly used biocompatible material, its basic properties are described briefly in the paper. It is recommended to provide a more detailed introduction to Parylene-C, including its physicochemical properties and the reasons for selecting it as the encapsulation layer.

Response: We sincerely appreciate the reviewer’s suggestions, which have significantly improved the quality of our manuscript. In response, we have added a brief explanation in the manuscript to highlight the physicochemical advantages of Parylene-C and the rationale for selecting it as the encapsulation material.

Revised in 162-168:

  1. The text mentions that grid structures enhance hydrophobicity while circular arrays enhance hydrophilicity, but it does not delve into the reasons behind these differences in wettability. It is recommended to include an analysis of the microscopic morphology of these structures and their impact on surface energy in the discussion section to help readers understand the underlying mechanisms.

Response: We sincerely appreciate the reviewer’s reminder. We acknowledge that the discussion on the physical mechanisms in this paper was insufficient. To address this shortcoming, we have added an in-depth discussion on this topic in the discussion section, aiming to enhance the overall value of the paper.

Revised in 303-319:

  1. The current cytotoxicity testing is based on a 24-hour MTT experiment. It is suggested to include longer-term cytotoxicity assessments (such as 7 days or 14 days) to verify the safety of the material over extended use.

Response: This test follows the ISO 10993-5 & 12 guidelines for biocompatibility testing. According to these standards, contact and extraction times of 24~72 hours at 37 ± 1°C are all considered acceptable for cytotoxicity testing. Since this experiment was conducted as a paid outsourced service, we sincerely apologize for not extending the testing duration to 7–14 days as suggested. In fact, as the reviewer correctly pointed out, a 24-hour test may not fully reflect the cytotoxic response of the biochip in its target cellular applications. However, the purpose of this paper is to preliminarily evaluate the biocompatibility of the modified encapsulation material. The long-term safety of the material will need to be determined based on the total dynamic culture duration for the specific target cells—potentially spanning from several weeks to month—which we are not yet able to establish at this stage. This study primarily aims to demonstrate the feasibility of biocompatible encapsulation and rapid localized surface modification. We kindly ask for your understanding regarding the preliminary nature of our current findings. Thank you very much for your valuable comments.

  1. L929 cells are a commonly used model for cytotoxicity testing, but to better simulate actual application environments, it is recommended to test with additional cell types (such as kidney cells or other organ-specific cells) to ensure the material's broad applicability.

Response: We appreciate the reviewer’s suggestions. Generally, cytotoxicity testing can be conducted using two approaches: one based on ISO standard cells and the other using specific cultured cells. In this study, we chose to follow ISO 10993-5 (in vitro cytotoxicity testing) to ensure compliance with regulatory requirements. Therefore, we selected L929 mouse fibroblast cells for the test. This approach has both advantages and limitations. The ISO standard is internationally recognized, providing robust evidence of safety and regulatory compliance. However, as the reviewer mentioned, it may not fully reflect the actual cytotoxicity in specific target cells (e.g., kidney cells). We selected this method considering the potential applicability of this study to various organ-on-a-chip platforms. The kidney organ-on-a-chip mentioned in the introduction serves only as an example to illustrate the need for sensor encapsulation technologies. Looking ahead, as multiple organ-on-a-chip systems (e.g., heart, liver, and kidney) may eventually be interconnected, a general biocompatibility standard without cytotoxicity is essential. Balancing these considerations, we have decided to temporarily adopt the ISO standard procedure for biocompatibility verification in this study. We have added supplementary explanations in the manuscript. Thank you again for your valuable comments.

Revised in 324-338:

  1. Some important works on the use of sensors in organ-on-a-chip devices need to be cited such as:
  • Optical sensor: Guo Z, Yang CT, Chien CC, Selth LA, Bagnaninchi PO, Thierry B. Optical Cellular Micromotion: A New Paradigm to Measure Tumor Cells Invasion within Gels Mimicking the 3D Tumor Environments. Small Methods. 2022 e2200471. doi: 10.1002/smtd.202200471.
  • Electrical sensor: Aydogmus, H., Hu, M., Ivancevic, L. et al. An organ-on-chip device with integrated charge sensors and recording microelectrodes. Sci Rep 13, 8062 (2023). https://doi.org/10.1038/s41598-023-34786-5

Response: Thank you for your suggestion. We have added these significant works to the references section of the manuscript.

Revised in 452-453, 484-485:

  1. Some paragraphs contain complex language. It is suggested to simplify the expression to make the article more readable. For example, long sentences can be reduced, and simpler vocabulary can be used to convey information.

Response: Thank you for your suggestion. Due to differences in language proficiency, we find it challenging to fully implement your recommendation. We sincerely apologize for not being able to clearly understand your intention. If you could kindly provide more specific instructions regarding the sections that need revision, we will make the necessary adjustments accordingly. Thank you.

  1. The symbols and units used in the text are sometimes inconsistent (e.g., the unit for contact angle is sometimes "°" and sometimes omitted). It is suggested to standardize the use of symbols and units throughout the text to ensure consistency.

Response: We have conducted a thorough review and made revisions based on your suggestions. Thank you for helping to improve the quality of our manuscript.

Reviewer 2 Report

Comments and Suggestions for Authors

The manuscript demonstrates a promising tool for mimicking glomerular filtration barrier. This kidney-on-a-chip device shows a potential for measuring fluid pressure exerted on the cells and incorporates a non-toxic scaffold to support cell attachment and cultivation. The concept of the device combines engineering and biological technologies to encapsulate the microenvironment for co-culture the glomerular endothelial cells and podocytes.

The manuscript lacks critical experiments to support the idea and requires major revisions. First, the manuscript demonstrates the feasibility of modifying the surface properties of Parylene-C, which is validated using the MTT assay. However, the experimental section does not include any demonstration of cell culture within the device, which is a key innovation proposed in this manuscript. The MTT assay on cells cultivated in 96-well plate instead of the device did not support the main idea of the manuscript.

Second, the cell line used in this manuscript is neither endothelial nor podocytes. Without inclusion of glomerular cells, the device cannot demonstrate the intended function of mimicking glomerular filtration barrier. Mouse/human immortalized endothelial or podocytes cell line need to be used.

Third, the authors repeatedly demonstrate the modification of the surface structure but do not mention or conduct any experiments related to how this structure impacts cell morphology or how cells adhere to the surface under specific fluid pressure conditions. Additional experiments and assays, such as western blotting and qPCR should be included.

Fourth, the device concept includes the filtration function of the porous structure, which is mentioned in Section 2 for observing albumin leakage. However, no additional experiments or results support this function of the device. Additional experiments, including albumin permeability assays with and without cultured cells, need to be evaluated.

Author Response

The manuscript demonstrates a promising tool for mimicking glomerular filtration barrier. This kidney-on-a-chip device shows a potential for measuring fluid pressure exerted on the cells and incorporates a non-toxic scaffold to support cell attachment and cultivation. The concept of the device combines engineering and biological technologies to encapsulate the microenvironment for co-culture the glomerular endothelial cells and podocytes.

  1. The manuscript lacks critical experiments to support the idea and requires major revisions. First, the manuscript demonstrates the feasibility of modifying the surface properties of Parylene-C, which is validated using the MTT assay. However, the experimental section does not include any demonstration of cell culture within the device, which is a key innovation proposed in this manuscript. The MTT assay on cells cultivated in 96-well plate instead of the device did not support the main idea of the manuscript.

Response: We appreciate the reviewer’s comments. Please allow me to clarify that the innovative aspects of this paper lie in the microengineering fabrication technologies, including biocompatible encapsulation and laser-localized wettability control. Cell culture is not the core focus of this study. The organ-on-a-chip serves as the implementation platform for the development of these technologies, providing the requirements and context for their application. Generally, cytotoxicity testing can be conducted using two approaches: one based on ISO standard cells and the other using specific cultured cells. In this study, we chose to follow ISO 10993-5 (in vitro cytotoxicity testing) to ensure compliance with regulatory requirements. Therefore, we selected L929 mouse fibroblast cells for the test. This approach has both advantages and limitations. The ISO standard is internationally recognized, providing robust evidence of safety and regulatory compliance. However, as the reviewer has rightly pointed out, it may not fully reflect the actual cytotoxicity in specific target cells (e.g., kidney cells). We selected the standard method considering the potential applicability of this study to various organ-on-a-chip platforms. The kidney organ-on-a-chip mentioned in the introduction serves only as an example to illustrate the need for sensor encapsulation technologies. Looking ahead, as multiple organ-on-a-chip systems (e.g., heart, liver, and kidney) may eventually be interconnected, a general biocompatibility standard without cytotoxicity is essential. Balancing these considerations, we have decided to temporarily adopt the ISO standard procedure for biocompatibility verification in this study. We have added supplementary explanations in the manuscript. We sincerely hope you can understand and accept our explanation.

Revised in 324-338:

  1. Second, the cell line used in this manuscript is neither endothelial nor podocytes. Without inclusion of glomerular cells, the device cannot demonstrate the intended function of mimicking glomerular filtration barrier. Mouse/human immortalized endothelial or podocytes cell line need to be used.

Response: Thank you very much for your valuable comments. The primary objective of this paper is to develop biocompatible encapsulation technology. Our research focuses on sensor and chip encapsulation techniques, rapid surface modification of encapsulation materials, and surface energy wettability control. The kidney organ-on-a-chip serves as the implementation platform for this study and is mentioned in the introduction to highlight the need for on-site sensors and the development of biocompatible encapsulation technologies. At this stage, we have not yet completed a fully developed kidney organ-on-a-chip. While the reviewer’s insights and suggestions are highly relevant and significant, they are beyond the scope of the results presented in this paper. In fact, we anticipate completing the kidney organ-on-a-chip, including the simulation of the glomerular filtration barrier function as mentioned, within approximately six months. However, that work will be the focus of a separate paper dedicated specifically to the organ-on-a-chip. At that time, the encapsulation technologies for sensors and chips developed in this paper will be utilized, but their engineering details will not be revisited. We sincerely hope this explanation provides clarity and is acceptable to you. Thank you again for your thoughtful and constructive comments.

  1. Third, the authors repeatedly demonstrate the modification of the surface structure but do not mention or conduct any experiments related to how this structure impacts cell morphology or how cells adhere to the surface under specific fluid pressure conditions. Additional experiments and assays, such as western blotting and qPCR should be included.

Response: Cell adhesion to the surface of the culture chamber under specific fluid pressure conditions is indeed a critical topic. However, this study focuses on interface engineering techniques for microchips.  and we would like to clarify that the paper does not intend to specifically address biological culture conditions. The primary objective of this work is to demonstrate how desired wettability can be freely achieved in different regions (various interfaces) of the chip. This goal is aimed at ensuring consistent or tailored fluidic conditions across different areas of the chip, such as microchannels, sensors, and culture chambers. This is accomplished through laser processing, enabling high-speed and low-cost localized modifications. The “cell adhesion” mentioned in your comments is generally achieved through parameters such as flow rate, pressure, and culture medium composition, which fall under the scope of biological culturing techniques. While we greatly appreciate your thoughtful reminder of the importance of cell culture, we regret to inform you that this aspect has not yet been addressed in our current work. We kindly ask for your understanding and hope to further explore and elaborate on this topic in a future paper dedicated to biological culture studies. This paper focuses exclusively on the engineering techniques for microchip fabrication processes. Thank you very much for your valuable comments.

  1. Fourth, the device concept includes the filtration function of the porous structure, which is mentioned in Section 2 for observing albumin leakage. However, no additional experiments or results support this function of the device. Additional experiments, including albumin permeability assays with and without cultured cells, need to be evaluated.

Response: Thank you for your comments. I believe the relevant explanations have already been provided earlier. This paper focuses on the development of biocompatible encapsulation and interface engineering techniques driven by the requirements of organ-on-a-chip systems. However, this study does not include the demonstration or investigation of biological functionality experiments and validations involving organ cells. In fact, the albumin permeability test you mentioned has already been conducted and successfully completed under static culture conditions. Nevertheless, implementing this test under dynamic culture conditions still presents several challenges that need to be addressed. We agree that this is an important topic requiring further experiments. We kindly ask for your understanding that this paper does not intend to disclose biologically related results at this stage. Instead, we aim for this work to make a significant contribution to the advancement of microengineering fabrication technologies. To enhance the quality of the paper, particularly in the discussion of the physical mechanisms, we have revised and added relevant content. We hope you find these updates valuable. We sincerely appreciate your valuable insights.

Revised in 303-319:

Round 2

Reviewer 2 Report

Comments and Suggestions for Authors

Accept to publish